# The Endogenous RIG-I Ligand Is Generated in Influenza A-Virus Infected Cells

**DOI:** 10.3390/v13081564

**Published:** 2021-08-07

**Authors:** Julia Steinberg, Timo Wadenpohl, Stephanie Jung

**Affiliations:** Institute of Cardiovascular Immunology, University Hospital Bonn, University of Bonn, 53127 Bonn, Germany; julia2.steinberg@ukbonn.de (J.S.); timo.wadenpohl@ukbonn.de (T.W.)

**Keywords:** RIG-I, endogenous RIG-I ligand, Influenza A-virus, vesicular stomatitis virus

## Abstract

As a result of a viral infection, viral genomes are not only recognized by RIG-I, but also lead to the activation of RNase L, which cleaves cellular RNA to generate the endogenous RIG-I ligand (eRL). The eRL was previously identified as a specific sequence derived from the internal transcribed spacer region 2, which bears a 2′3′ cyclic phosphate instead of the common 5′ triphosphate. By now, the generation of the eRL and its immunostimulatory effect were shown both in vitro and in reporter systems. In this work, we aimed to elucidate whether the eRL is also generated in Influenza A (IAV) and vesicular stomatitis virus (VSV) infected cells. RNA was extracted from virus-infected cells and used for immunostimulations as well as specific PCR-strategies to detect eRL cleavage. We show that the eRL is generated in IAV infected HEK293 cells, but we could not detect specific eRL fragments in VSV infected cells. Further, RIG-I mediated IFN-response depends not only on viral genomes but also on the eRL, as immunostimulatory properties remain present under 5′triphosphate degrading conditions. In summary, we prove the IAV infection induced eRL generation in HEK293 cells, amplifying the innate immune response.

## 1. Introduction

The innate immune system can recognize infections and cell damage using germline-encoded pattern recognition receptors (PRRs) [1,2]. These receptors detect so-called pathogen-associated molecular patterns (PAMPs), which are characteristic for a specific group of pathogens. Binding of a PAMP by its specific PRRs leads to signal transduction and interferon (IFN) production, eventually resulting in the expression of interferon-stimulated genes (ISGs).

Regarding these pathways, a key role is played by the family of cytoplasmic RIG-I like receptors (RLRs), which comprise the activating receptors Retinoic Acid Inducible Gene I (RIG-I) and Melanoma Differentiation Associated Gene 5 (MDA5) just as the accessory receptor Laboratory of Genetics and Physiology *2* (LGP2) [3]. Whereas MDA5 recognizes longer and more complex RNA structures, the classical ligand of the eponymous receptor RIG-I is at least partially double-stranded RNA or hairpin structures of viral genomes with a 5′ tri- or diphosphate [4,5,6,7,8]. However, not only viral RNA can activate RIG-I, but also cellular RNA, leading to an amplification of the danger signal in viral infections [9]. In this case, the recognition of double-stranded RNA by the oligoadenylate synthetase leads to downstream activation of RNase L. The latter cleaves cellular RNA, which in turn functions as an endogenous RIG-I ligand (eRL). We identified the eRL as a specific sequence derived from the internal transcribed spacer 2 (ITS2)-region of 45S ribosomal RNA (rRNA) [10]. Unlike the classical RIG-I ligand, the eRL is not phosphorylated at the 5′ end, but its immunostimulatory activity depends on a 2′3′ cyclic phosphate and GTP binding by internal G-quadruplex structures [10].

Influenza A virus (IAV) infections are a serious health problem, regularly causing seasonal outbreaks and accounting for approximately 250,000 to 300,000 deaths per year [11,12,13]. In the viral lifecycle, the segmented negative-stranded RNA genome is released from the endosomal compartment into the cytoplasm before it is imported into the nucleus for viral transcription and replication. During this process, IAV infection is detected by RIG-I, which binds the panhandle structures of the viral genomes [7,14].

Similar to IAV, vesicular stomatitis virus (VSV) infection is also recognized by RIG-I binding of VSV genomes [15]. Due to its ability to infect a wide range of host cells and the fact that it is non-pathogenic for humans, VSV is broadly used as a model virus [16,17]. VSV has been reported to rapidly induce apoptosis in the infected cell via both the extrinsic and the intrinsic pathways, with the VSV-matrix protein playing a crucial role [17,18].

Using both IAV and VSV as a model, we aimed to investigate if, and under which conditions, the eRL is generated in viral infections. We show that the eRL is generated in IAV infections, but we were unable to detect eRL cleavage in VSV-infected cells under the conditions we used. Furthermore, we prove that the eRL confers immunostimulatory properties to RNA from virus-infected cells. In summary, we contributed further insights into how cellular RNA ligands can amplify danger signaling in viral infection.

## 2. Materials and Methods

### 2.1. Kits and Reagents

Transfection reagent Lipofectamine 2000 was obtained from ThermoFisher (Cat. No. 11668019, Waltham, MA, USA). Opti-MEM was provided by ThermoFisher (Cat. No. 31985070). Poly I:C was obtained from GE Healthcare (Cat. No. 27473201, Little Chalfont, UK). 5′ppp-RNA as RIG-I ligand was synthesized using the T7 RNA synthesis kit from NEB (Cat. No. E2040S, Ipswich, MA, USA). eRL was produced as described [10]. Human IFN-α was quantified using the IFN alpha Human Matched Antibody Pair of eBiosciences (Cat. No. BMS216MST, San Diego, CA, USA). IFN-β was measured using Human IFN-beta DuoSet ELISA (Cat. No. DY814-05, R&D Systems, Minneapolis, MN, USA). Ficoll was provided from GE Healthcare (Cat.No. 17-1440-02). Cell viability was controlled using the Cell Counting Kit 8 (WST-8/CCK8) (Cat. No. ab228554, Abcam, Cambridge, UK).

### 2.2. Cells and Viruses

HEK293T cells were grown in DMEM (Cat. No. 41965062, ThermoFisher) with 10% FCS (Cat. No. 10438026), 2 mM L-glutamine (Cat. No. 25030024), 100 units/mL penicillin and 100 µg/mL streptomycin (Cat. No. 15140122). HEK293-RIG-I-IFN-β reporter cells (HEK-RIG-I) were cultivated in DMEM with 10% FCS, 2 mM L-glutamine, 100 units/mL penicillin, 100 µg/mL streptomycin and 0.7 mg/mL Geneticin (Cat. No. 10131035) as selection marker. Vero and MDCK cells were grown in RPMI (Cat. No. 21875091, ThermoFisher) with 5% FCS, 2 mM L-glutamine, 100 units/mL penicillin and 100 µg/mL streptomycin. PBMCs were purified from blood of healthy human donors using Ficoll gradient centrifugation and were resuspended in RPMI with 2 mM L-glutamine, 100 units/mL penicillin, 100 µg/mL streptomycin, non-essential amino acids (Cat. No. 1114035), sodium pyruvate (Cat. No. 11360039) and 2% serum of AB positive male donors (Cat. No. H4522, Merck, Darmstadt, Germany). IAV (Influenza A/Puerto Rico/8/1934) was grown and harvested from 9-day-old embryonated chicken eggs [19]. At day 1 after IAV inoculation, eggs were chilled overnight at 4 °C prior to harvesting of allantoic fluid. VSV was produced in BSR T7 cells cultured in DMEM with 10% FCS, 2 mM L-glutamine, 100 units/mL penicillin and 100 µg/mL streptomycin. Viral titers were determined as plaque-forming units (PFU) per ml in a plaque assay using MDCK cells for IAV and Vero cells for VSV. Cell monolayers were infected with the respective virus at 10-fold dilutions for 1 h. Afterwards, inoculum was removed, and cells were overlaid with MEM (Cat. No. 31095029, ThermoFisher), 2% FCS, 1.5% sodium carboxymethyl cellulose (Cat. No. 419273, Merck), 30 mM HEPES (Cat. No. 15630080, ThermoFisher) and 100 units/mL penicillin and 100 µg/mL streptomycin.

### 2.3. Light Microscopy

Microscopy was performed using a CKX53 microscope with a 10× CAchN objective (Olympus, Tokyo, Japan). Pictures were taken using a LC30 camera and the software cellSense Entry (Version 2.1, Olympus).

### 2.4. Luciferase Assay

HEK-RIG-I cells were seeded with a density of 3 × 10^4^ cells/well in 96-well plates one day prior to immunostimulation. For stimulations, 3 µg/mL Poly I:C or either 2 µg/mL 5′ppp-RNA, eRL or RNA of (non-)infected cells were used as indicated. For IAV RNA, 0.08 µg/mL RNA were added to 2 µg/mL RNA of uninfected HEK293T cells prior to stimulation. Stimuli were prepared in RNase-free water in a total volume of 20 µL. CIP treatment was performed by incubation at 37 °C for 30 min after addition of 2 µL CutSmart buffer and 1 µL QuickCIP (Cat. No. M0525S, NEB). In case of hydrochloric acid treatment, 2 µL 100 mM HCl (Cat. No. 131020.1611, AppliChem, Darmstadt, Germany) were added to each sample, followed by a 30 min incubation at 37 °C and neutralization with 2 µL 100 mM NaOH (Cat. No. 1091371000, Merck). Stimuli were mixed with 0.5 µL Lipofectamine 2000 and 144.5 µL Opti-MEM (Cat. No. 31985062, Thermo-Fisher) and after 5 min incubation at room temperature, 50 µL/well were used for stimulation in triplicates. Then, 20 h after stimulation, cells were lysed in Reporter Lysis Buffer (Cat. No. E4550, Promega, Madison, WI, USA) and firefly luciferase activity was measured using the TriStar^2^ microplate reader by Berthold Technologies (Bad Wildbad, Germany). Fold IFN-β induction was normalized to mock-treated cells.

### 2.5. PBMC Stimulation

PBMCs were seeded at 3 × 10^5^ cells/well in a 96-well plate in 100 µL growth medium. PBMCs were stimulated with RNA ligands (see above) at 0.08 µg/mL. RNA was diluted in 25 µL Opti-MEM per 96-well and combined with 0.5 µL Lipofectamine in 25 µL Opti-MEM. After incubation at room temperature for 5 min, 50 µL PBMC medium without serum were added and RNA solution was used for immune stimulation. Then, 20 h post stimulation, supernatants were harvested and subjected to cytokine measurement.

### 2.6. RNA-Isolation and qRT-PCR

RNA was isolated using Trizol (Cat. No. 15596018, Invitrogen, Carlsbad, CA, USA) according to the manufacturer’s instructions. DNA contaminations were removed using DNase I by Roche (Cat. No. 4716728001, Basel, Switzerland). The ratio of eRL to uncut ITS2 was determined as described previously [10] using QuantiTect Reverse Transcription Kit (Cat. No. 205311, Qiagen, Hilden, Germany) and DreamTaq Green PCR Master Mix (2X) (Cat. No. K1081, Thermo Scientific). Band intensity was measured using GelQuantNET (omicX, Le Petit-Quevilly, France).

### 2.7. RNase A Digestion

For complete digestion of HEK293T cell derived RNA, RNase A (Cat. No. 19101, Qiagen, Hilden, Germany) was applied at 2.5 × 10^−5^ U/µg RNA. Digestion was performed in 1× HEPES buffered saline (Cat. No. 51558, Merck) for 1 h at 37 °C.

### 2.8. Viral Infections

HEK293T cells were grown to 80% confluence and infected with IAV and VSV at a multiplicity of infection (MOI) of 0.75, 0.075 and 0.0075 plaque-forming unit (PFU)/cell in growth medium without FCS. Then, 1 h post infection (p.i.), 2 infection volumes of growth medium were added, and cells were incubated at 37 °C with 5% CO_2_. Influenza infected cells were harvested 24 h p.i.; VSV infected cells were harvested 12–16 h p.i.

### 2.9. Cell Viability Assay

HEK293T cells were seeded in a density of 3 × 10^4^ cells/well in 96-well plates one day prior to viral infection or immunostimulation according to the previously described procedure (refer Viral Infections and Luciferase Assay). Then, 24 h after infection resp. 20 h after immunostimulation, 10 µL WST-8 substrate were added to each well and absorbance was measured at 450 nm after 4 h of incubation at 37 °C. Viability was normalized to moc- treated cells.

### 2.10. Western Blotting

Virus infected HEK293T cells were lysed 12 resp. 24 h p.i. in lysis buffer (50 mM Tris/HCl pH 8.0, 150 mM NaCl, 1% NP-40), cell debris was removed by centrifugation at 15,000× *g* for 15 min and protein concentration was analyzed via Bradford assay. Then, 5 µg total protein were separated on a 4–20% SDS gel and used for Western blotting. For VSV detection, the anti-VSV-G antibody [8G5F11] (Cat. No. EB0010, Kerafast, Boston, MA, USA) and anti-mouse IgG, HRP-linked antibody (Cat. No. 7076, Cell Signaling Technology, Danvers, MA, USA) were used. IAV protein expression was detected by use of Influenza A m2 [14C2] HRP antibody (Cat. No. sc-32238 HRP, Santa Cruz Biotechnology, Santa Cruz, CA, USA). GAPDH was detected using GAPDH (14C10) Rabbit mAb (HRP Conjugate) (Cat. No. 3683S, Cell Signaling Technology).

### 2.11. Statistical Analysis

Experiments were performed three to six times. Data were analyzed for normality using Kolmogorov–Smirnov and D’Agostino–Pearson tests and are represented as arithmetic means ± SD. If not stated otherwise, statistical analysis of normally distributed data was based on paired two-tailed t tests. In this way, the immunological effect of differently treated matched samples (e.g., infected vs. uninfected cells from an individual experiment, with a total number of 3 to 6 independent experiments) was tested. If there was no normal distribution, data were analyzed using Wilcoxon test (indicated in the figure legend).

## 3. Results

### 3.1. eRL Is Generated in IAV-Infected Cells

We aimed to investigate if the eRL is generated in viral infections, and how eRL production correlates with cell viability. HEK293T cells were infected with both IAV and VSV and infection efficiency was demonstrated by Western blotting targeting viral structural proteins (Figure A1c,f). Whereas VSV infection seems to induce vigorous cytopathic effects resulting in decreased cell density and decreased cell viability (Figure A1d,e), IAV infection seems less cytotoxic at indicated MOIs (Figure A1a,b) resulting in plaque formation but no detectable decrease of cell viability.

In order to monitor eRL generation in virus-infected cells, we employed a previously established PCR-strategy distinguishing between eRL and uncut ITS2-RNA (Figure 1a and [10]). Applying this method, two forward (fw)-primers are used: if the longer, uncut RNA sequence is present and the fw-uncut primer can bind, this leads to displacement of the fw-eRL primer, resulting in preferential amplification of the uncut template (Figure 1a, right panel, mock sample). Generation of the eRL leads exclusively to synthesis of the shorter sequence (Figure 1a, right panel, digested sample). In the case of intracellular eRL generation, the ratio between uncut sequences and eRL sequences is consequently reversed (compare Figure 1b, line H). IAV infection resulted in clearly detectable eRL induction at an MOI of 0.75 PFU/cell (Figure 1b,c). In contrast, VSV infection did not induce a detectable eRL generation at applied MOIs (Figure 1d,e). Consequently, we mainly focused on IAV at an MOI of 0.75 PFU/cell, for further experiments.

### 3.2. Virus-Induced RNA Species Stimulate Type I IFN Via RIG-I

After verifying eRL generation in response to IAV infection, we tested immunostimulatory activity of virus-induced RNA species. RNA extracted from IAV-infected cells or mock treated cells was used to stimulate PBMCs, then IFN-α and IFN-β release was determined in ELISA (Figure 2a,b). In contrast to the RNA extracted from mock-treated cells, RNA derived from IAV-infected cells induced a type I IFN response in human PBMCs.

To verify RIG-I specificity of IAV-induced RNA species, RNA from IAV infected or mock-treated cells was employed for the stimulation of HEK293-RIG-I-IFN-β reporter cells (Figure 2c). These cells ectopically express RIG-I and firefly luciferase controlled by an IFN-β promotor, thus RIG-I activation leads to luciferase expression. Treatment with RNA from virus infected cells resulted in IFN-β induction. In addition, RNA extracted from VSV-infected cells induced a Type-I IFN response in PBMCs and HEK RIG-I cells (Figure A2). We concluded that both viral infections induce immunostimulatory RNA species which stimulate RIG-I.

### 3.3. IAV-Induced RNA Species Activate Type I IFN Release Independently of IAV Genomes

In the further course of this study, we examined whether IAV-induced RNA species exhibit properties that are characteristic for the eRL. Whereas calf intestine phosphatase (CIP) removes 5′ phosphorylation of classical RIG-I ligands, such as IAV genomes, the 2′3′ cyclic phosphate of the eRL is resistant to CIP treatment [20]. Indeed, dephosphorylation of 5′ppp-RNA resulted in a complete loss of immunostimulatory capacity (Figure 3a). In contrast, both Poly I:C (Figure 3b) and eRL (Figure 3c), not depending on 5′ppp, were still immunostimulatory after CIP treatment, though CIP-treatment reduced the immunostimulatory potential of these RNA species. To show that the 5′ phosphates of IAV genomic RNA are fully accessible to CIP-mediated dephosphorylation, we spiked 2 µg/mL non-stimulatory RNA with 0.4 µg/mL, 0.08 µg/mL, 0.016 µg/mL and 0.003 µg/mL IAV genomes (Figure A3a), or 0.08 µg/mL IAV genomes (Figure A3b,c). In both cases, RNA was subjected to CIP treatment prior to immunostimulation of PBMCs (Figure A2a,b) or HEK reporter cells (Figure A3c), which completely abolished immunostimulation by IAV genomes. However, RNA from IAV infected cells still retained immune activating potential after CIP treatment (Figure 3d). Although eRL generation was not detectable in VSV infection, VSV-induced RNA also showed similar characteristics (Figure A4). We concluded that immunostimulation by RNA generated in infected cells depends not only on viral genomes, but also on other RNA species.

### 3.4. Immunostimulation by IAV-Induced RNA Depends on 2′3′ Cyclic Phosphate

To elucidate whether IAV-induced RNA stimulates type I IFN release due to eRL activity, we focused on 2′3′ cyclic phosphate which is crucial for eRL function. Treatment with low HCl concentrations was previously shown to hydrolyze 2′3′ cyclic phosphate structures without affecting RNA integrity [21,22,23]. Indeed, HCl treatment abolished the immunostimulatory potential of eRL in HEK293-RIG-I-IFN-β reporter cells (Figure 4a), whereas both 5′ppp-RNA and Poly I:C, not depending on cyclic phosphate, were still immunostimulatory after HCl treatment (Figure 4b,c). However, IAV-induced RNA species were not immunostimulatory anymore after HCl treatment (Figure 4d). Consequently, IAV infection of HEK 293T cells induce immunostimulatory features which are identical to the eRL. Though VSV infection did not induce detectable amounts of eRL cleavage, immunostimulation by VSV-induced RNA species was also sensitive to HCl treatment (Figure A5). Cell viability was not affected by HCl treatment of RNA ligands, as HCl was neutralized with NaOH prior to immunostimulation (Figure A6).

## 4. Discussion

RIG-I activation by endogenous RNA fragments has been reported in several studies [9,10,24]. However, the origin and structure of the eRL has remained enigmatic for a long time. We were the first to show that the eRL is excised from the ITS2 region of 45S rRNA and depends on 2′3′ cyclic phosphate and GTP-binding structures for its immunostimulatory activity, but did not include viral infections in this previous report [10]. Therefore, we aimed at investigating if the eRL is generated in viral infections and to what extent it contributes to RIG-I activation. Indeed, we show that IAV-infection leads to eRL cleavage, which amplifies the Type I-IFN-response to IAV RNA (Figure 5).

First, we studied the correlation of eRL generation with impaired cell viability, to rule out any unspecific effect of apoptosis-induced RNA fragmentation. Of note, VSV infection profoundly induced cell death, but no eRL synthesis was detectable by PCR (Figure A1e, Figure 1d,e). In contrast, IAV infection strongly induced eRL fragments at an MOI of 0.75 PFU/cell (Figure 1b,c) after 24 h p.i. At the same time, plaque formation was induced but cell viability was not detectably decreased (Figure A1a,b), indicating that RNase L activity and eRL biogenesis are not the result of apoptosis [25]. Consequently, we conclude that eRL cleavage is specifically induced in IAV infection. We suspect that also lower MOI might induce lower levels of eRL generation, which are not detectable due to the sensitivity of available assays.

Next, we aimed at elucidating the physiological impact of IAV-induced eRL synthesis. Applying PBMC and RIG-I dependent reporter cells, we showed that IAV infection induces cellular RNA species activating RIG-I (Figure 2). Hereupon we showed with different treatment modalities that this immune activation does not exclusively depend on IAV genomes but also on other RNA species of cellular origin. Unlike the classical RIG-I ligand depending on a 5′ phosphorylation, which is sensitive to CIP treatment, eRL activity relies on a 2′3′ cyclic phosphate which is not removed by CIP [20]. Although CIP treatment decreased eRL-mediated IFN-β induction by about 50% and Poly I:C mediated IFN-β induction by 30%, which might be caused by the strong RNA-binding properties of CIP, IFN-β induction was clearly detectable (Figure 3b,c). However, RIG-I activation by IAV genomic RNA is known to depend on 5′ phosphates [7]. Indeed, CIP treatment of 5′ppp-RNA or of HEK293T-derived RNA spiked with excessive amounts of IAV genomes resulted in a complete loss of immunostimulation (Figure 3a, Figure A3). In contrast, immunostimulation by RNA extracted from IAV infected cells was still detectable (Figure 3d). The decrease of stimulatory effects in CIP-treated RNA compared to the untreated RNA reflects the dephosphorylation of IAV genomes. Consequently, immunostimulation in IAV-infected cells does not depend solely on 5′ppp RNA and thus does not depend solely on IAV genomes.

We further investigated whether immunostimulation in IAV-infected cells is mediated by eRL. We have previously shown that eRL immunostimulation depends on 2’3’ cyclic phosphate, which can be opened by HCl treatment [10]. Indeed, we observed that RNA from IAV-infected cells was also no longer immunostimulatory after HCl treatment (Figure 4). Consequently, immunostimulation via endogenous RIG-I ligands plays an important role in IAV-mediated RNA stimulation.

Although we could not detect eRL fragments in RNA derived from VSV-infected cells (Figure 1d,e), this RNA showed immunostimulatory properties similar to those of the eRL (Figure A2, Figure A4 and Figure A5). Though we cannot rule out that small undetectable amounts of eRL partly contribute to this effect, we rather suspect that endogenous RIG-I ligands other than the eRL sequence derived from the ITS2 region are mainly responsible for IFN induction [10]. In general, we consider the VSV data to be less robust than the IAV data because VSV, unlike IAV, exhibits high cytotoxicity at the concentrations used (Figure A1). Nevertheless, they support the assumption that properties of the eRL play an important role in immune recognition of viral infections.

We could prove that the previously described eRL is generated in IAV-infected cells, and we presume that it plays a key role in IAV-induced immunostimulation. However, we cannot state that this specific sequence is the only sequence contributing to RIG-I activation by self-RNA in IAV infection, as more eRL sequences could be involved. Regarding VSV infection, it seems conceivable that further eRL sequences are generated, leading to immune activation. For instance, the 28S rRNA expansion segment ES7L published by Lässig et al. could be involved in virus-induced eRL activity [26]. In addition, viral induction of a 5S rRNA pseudogene transcript and long non-coding RNA has been reported to contribute to self-RNA recognition by RIG-I [27,28].

Previous reports showed that influenza NS1 protein inhibits 2′-5′ oligoadenylate synthetase (OAS) activation by sequestering dsRNA [25,29]. As the eRL is generated in an RNase L and OAS-dependent manner [10], we suspect that this inhibition was only partial in our setup, perhaps due to low NS1 expression levels at an early timepoint of the infection. However, we did not detect any visible and non-specific IAV-induced RNA degradation in agarose gel electrophoresis either (data not shown). Thus, we hypothesize that in IAV infection, a small amount of dsRNA is recognized by OAS, leading to weak RNase L activation and eRL generation.

So far, the physiological function of the eRL has not been conclusively clarified as its generation depends on the OAS, which is an IFN-induced gene upregulated subsequently to RIG-I activation [30]. However, Malathi et al. postulated in 2007 that RNase L-generated RNA species amplify RIG-I activation in Sendai and Encephalomyocarditis virus infection, resulting in a positive feedback-loop [9]. Using hepatitis C virus as an example, this group further demonstrated that viral RNA also acquires RIG-I activating properties through RNase L activity [31].

There is growing evidence that activation of innate immunity by RNA degradation products is not restricted to viral infections, RNase L, or RIG-I. Cellular RNAs are processed, for example, by IRE-1 in the context of the unfolded protein response, whereupon they can activate RIG-I [24]. Since it has recently been shown that processing of bacterial RNA by RNase T2 is also essential for activation of Toll-like receptor 8, the importance of RNA degradation products in innate immunity is probably greater than expected [32,33]. This calls for further research in these areas to understand the defense against infections as well as potential autoimmune disease development.

## 5. Conclusions

Here we show that the eRL is generated in IAV-infected cells and plays a considerable role in IAV immunorecognition. In accordance with previously published data, RIG-I activation by IAV-induced eRL depends on 2′3′ cyclic phosphate and is not a result of cell death. The knowledge gained here will contribute to a better understanding of immune recognition of viral infections and resulting immunopathogenesis. Additional and broader studies will reveal the role of the eRL in other viral infections and how this correlates with the viral life cycle.

## Figures and Tables

**Figure 1 viruses-13-01564-f001:**
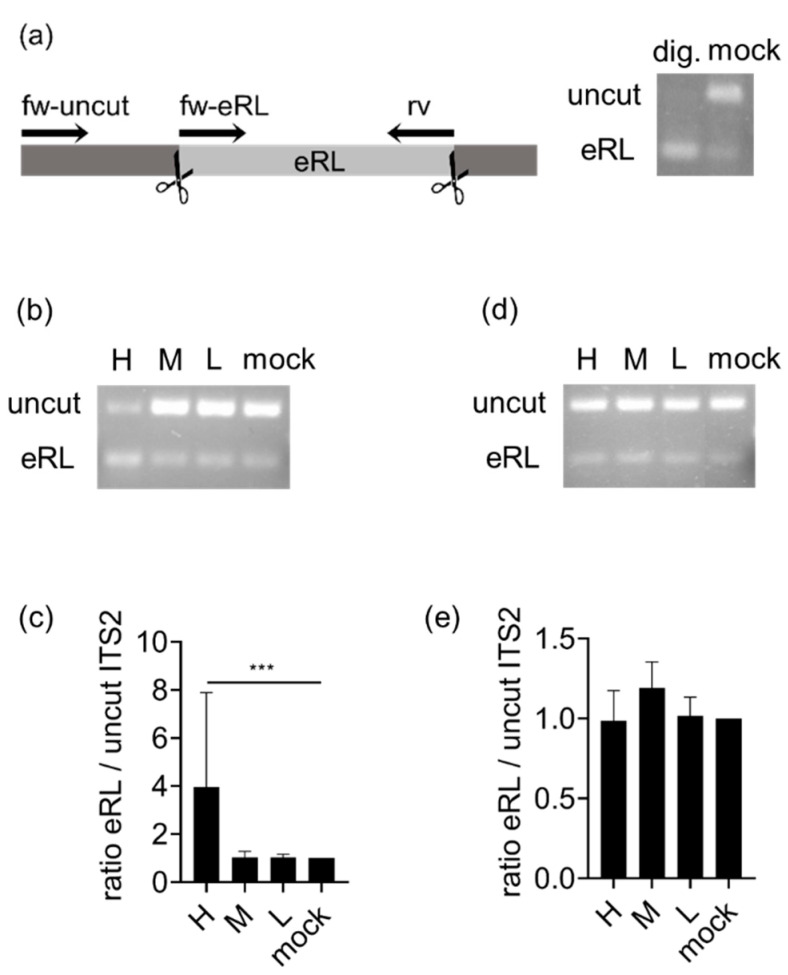
Measurement of eRL generation in infected cells. (**a**) Left panel: Schematic illustration of the PCR strategy to distinguish between eRL and uncut ITS2 fragments. Right panel: RNA digested with RNase A (dig.) or mock-treated RNA (mock) were subjected to the depicted PCR strategy and analyzed in agarose gel-electrophoresis. (**b**–**e**) HEK293T cells were infected with IAV (**b**,**c**) or VSV (**d**,**e**) in decreasing MOI (H ≙ 0.75 PFU/cell; M ≙ 0.075 PFU/cell; L ≙ 0.0075 PFU/cell). RNA was extracted and subjected to the PCR-strategy depicted in (a). (**b**,**d**) Representative fragment analysis in a 3% agarose gel stained with PeqGreen. (**c**) Graph combines three independent experiments each in technical quadruplicates (twelve measurements per data point ± SD). Data from IAV-infected cells were compared to mock-treated cells and analyzed using Wilcoxon-test. *** *p* < 0.001. (**e**) RNA was extracted from VSV-infected cells and analyzed for eRL generation. Graph combines three independent experiments, compared to mock-treated cells (three measurements per data point ± SD).

**Figure 2 viruses-13-01564-f002:**
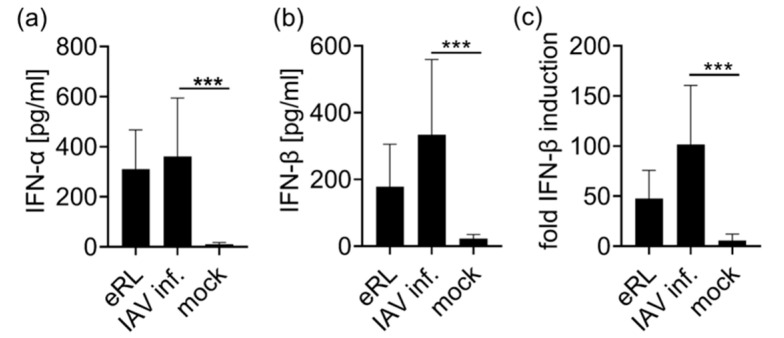
Immunostimulation by IAV-induced RNA species. (**a**,**b**) PBMCs were stimulated with eRL, RNA extracted from HEK 293T cells infected with IAV at an MOI of 0.75 PFU/cell or mock-treated cells, and cytokine levels of IFN-α (**a**) and IFN-β (**b**) were detected in ELISA. Graphs combine six independent experiments each in biological duplicates (twelve measurements per data point ± SD) and data of (**b**) were analyzed using Wilcoxon test. *** *p* < 0.001. (**c**) HEK293-RIG-I-IFN-β reporter cells were stimulated with eRL, RNA extracted from HEK293T cells infected with IAV at an MOI of 0.75 PFU/cell or mock-treated cells, and fold IFN-β induction was determined after 20 h. Graph combines three independent experiments each in biological triplicates (nine measurements per data point ± SD). *** *p* < 0.001.

**Figure 3 viruses-13-01564-f003:**
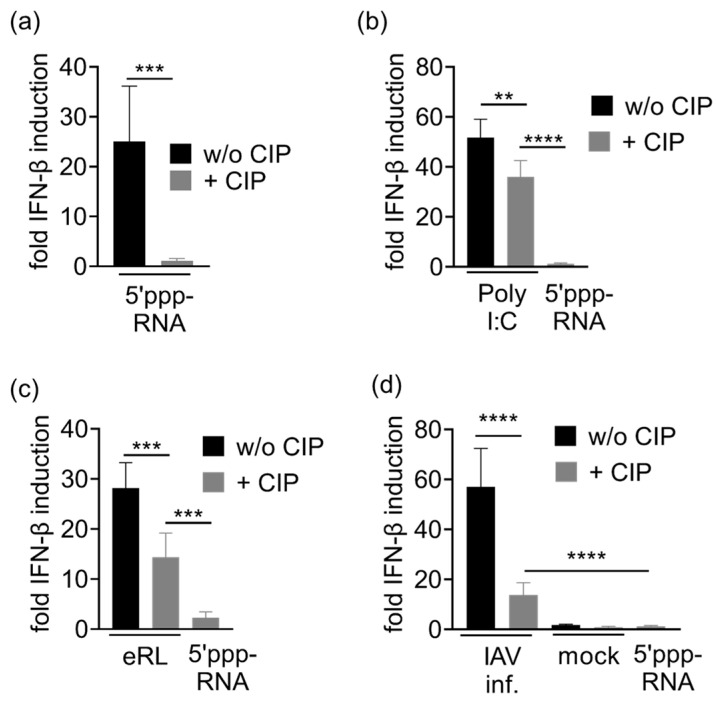
HEK293-RIG-I-IFN-β reporter cells were stimulated with RNA ligands with or without CIP treatment and fold IFN-β induction was determined after 20 h. Graphs depict three independent experiments each in biological triplicates (nine measurements per data point ± SD). (**a**) Stimulation with 5′ppp-RNA ± CIP. *** *p* < 0.001. (**b**) Stimulation with Poly I:C ± CIP and CIP-treated 5′ppp-RNA. ** *p* < 0.01, **** *p* < 0.0001. (**c**) Stimulation with eRL ± CIP and CIP-treated 5′ppp-RNA. *** *p* < 0.001. (**d**) Stimulation with RNA extracted from IAV-infected cells ± CIP, mock-infected cells ± CIP and CIP-treated 5′ppp-RNA. **** *p* < 0.0001.

**Figure 4 viruses-13-01564-f004:**
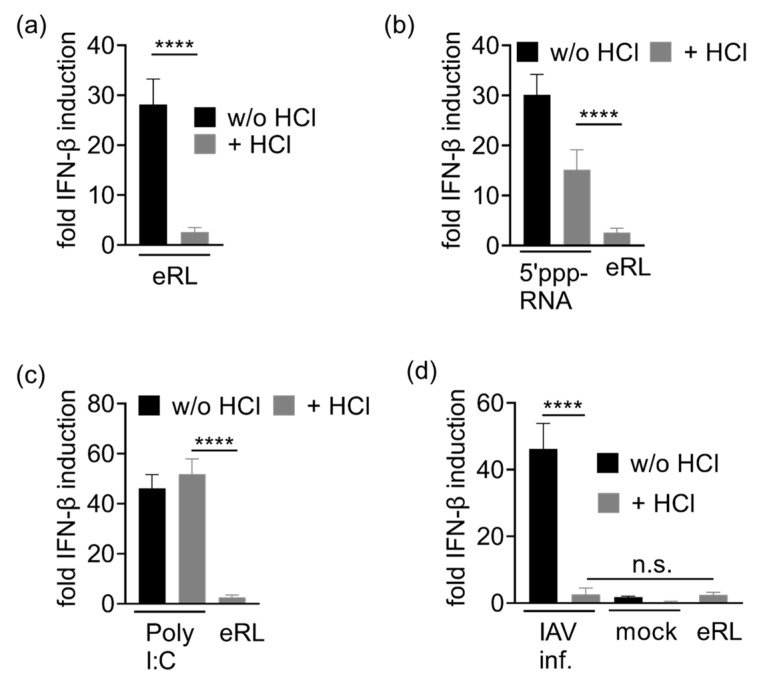
HEK293-RIG-I-IFN-β reporter cells were stimulated with RNA ligands with or without HCl treatment and fold IFN-β induction was determined after 20 h. Graphs depict three independent experiments each in biological triplicates (nine measurements per data point ± SD). **** *p* < 0.0001. (**a**) Stimulation with eRL ± HCl. (**b**) Stimulation with 5′ppp-RNA ± HCl and HCl-treated eRL. (**c**) Stimulation with Poly I:C ± HCl and HCl-treated eRL. (**d**) Stimulation with RNA extracted from IAV-infected cells ± HCl, mock-infected cells ± HCl and HCl-treated eRL.

**Figure 5 viruses-13-01564-f005:**
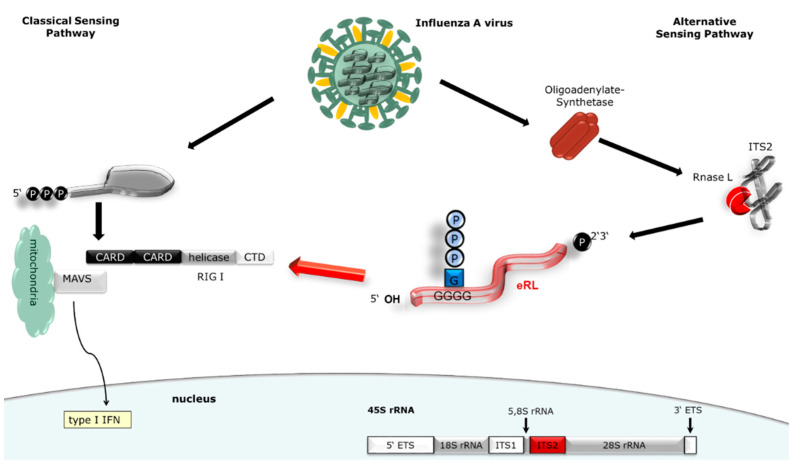
Graphical abstract of eRL induced RIG-I activation in viral infections. Viral genomes bearing a 5′ triphosphate are directly recognized by RIG-I. Additionally, RNase L activity leads to eRL generation, which is derived from the ITS2 region of 45S rRNA and in turn activates RIG-I. This immunostimulatory activity does not depend on 5′ phosphorylations but on a 2′3′ cyclic phosphate.

## Data Availability

Not applicable.

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
