# Peer review of "The Endogenous RIG-I Ligand Is Generated in Influenza A-Virus Infected Cells"

_viruses, 2021, doi:10.3390/v13081564_

Round 1
Reviewer 1 Report
In this manuscript titled "The endogenous RIG-I Ligand is generated in Influenza A-virus infected cells", Julia et al. found that endogenous RIG-I ligand (eRL) could be generated in the influenza A virus infected but not in the vesicular stomatitis virus infected HEK293 cells. They then found that RIG-I mediated IFN-response is depended not only on IAV genomes but also on the eRL. They authors follow up their previous study where they identified eRL in cells to demonstrate that eRL can be generated in virus-infected cells. In general, this study is of interest. However, there are many concerns need to be addressed.
Major concerns:
- In the section of Materials and Methods, catalog numbers are needed for reproducibility.
- Line 80, what kind of amino acids?
- Line 82, what kind of eggs, for how long? how do you culture the BSR T7 cells?
- How did you titer the virus? How did you culture the cells for virus titration?
- Line 87, Luciferase assay: does this assay have a normalizer? if not, how did you normalize the results?
- Line 135, please explain why did you choose paired over unpaired t-test.
- Line 140, virus infection efficiency cannot be controlled by blotting but only be demonstrated. Nevertheless, a housekeeping gene such as actin should be uses as internal control in western blot figure 1b and 1d.
- Line 141, cell viability cannot be controlled in light microscopy. Here, cytopathic effect can be observed by microscopy. The authors would need to use cell viability assay to demonstrate cell death.
- Figure 1 is not so important and can be put in supplemental data. Figure 1 should be something very important and striking. The current figure 1 is sort of disappointing. The authors should consider re-organize the figures.
- According to the illustration in figure 2A. fe-eRL plus rv primers can also amplify uncut template. Therefore, eRL band is not specifically derived from cut eRL. The authors should explain the specificity of this PCR system. Otherwise the data obtained from this assay is highly shaky.
- The legend of Figure 2 requires a topic/summary sentence.
- Figure 3b. eRL is missing. the author need to describe how they stimulated the cells with the RNAs. Is it transfection? How much RNAs? How long?
- Topic sentence if missing in the legend of Figure 3.
- The data from Figure 3 is basically well-known because it contains genomic RNA of IAV. There is no novelty in figure 3.
- Line 190. (CIP)removes -> (CIP) removes.
- Mock treatment controls are missing in figure 4.
- In figure 4, W/O CIP vs +CIP should be statistically analyzed and discussed.
- Figure 4d is the most valuable data in figure 4. However, mock-infected RNA is a required control to demonstrate the stimulus was derived from infection rather than background. VSV infected cell RNA should also be included in figure 4.
- Mock treatment controls are missing in figure 5 .
- VSV infected sample should also be used in fugue 5 to demonstrate no eGL was induced.
Author Response
Thank you very much for your positive and helpful feedback. We have processed the points as listed below. We think that the experiments you suggested greatly improved the quality of the publication, and we hope that you now find our work as compelling as we do.
Major concerns:
- In the section of Materials and Methods, catalog numbers are needed for reproducibility.
Thank you very much for this valuable suggestion. Catalog numbers have been integrated into the manuscript.
- Line 80, what kind of amino acids?
This information has been integrated into the M&M part.
- Line 82, what kind of eggs, for how long? how do you culture the BSR T7 cells?
Thank you very much for your helpful advices. This information has been integrated into the M&M part.
- How did you titer the virus? How did you culture the cells for virus titration?
This information has been integrated into the manuscript.
- Line 87, Luciferase assay: does this assay have a normalizer? if not, how did you normalize the results?
Thank you very much for your patience. Fold IFN-β induction was normalized to mock-treated cells.
- Line 135, please explain why did you choose paired over unpaired t-test.
We chose paired over unpaired t-test, since these respective samples belonged together and were analyzed in the same experiment. In this way, the immunological effect of differently treated matched samples (e.g. infected vs. uninfected cells from an individual experiment, with a total number of 3 to 6 independent experiments) was tested.
- Line 140, virus infection efficiency cannot be controlled by blotting but only be demonstrated. Nevertheless, a housekeeping gene such as actin should be uses as internal control in western blot figure 1b and 1d.
Thank you very much for this helpful comment. The sentence has been changed, and GAPDH as an internal control has been included in the new Supplementary Figure A1.
- Line 141, cell viability cannot be controlled in light microscopy. Here, cytopathic effect can be observed by microscopy. The authors would need to use cell viability assay to demonstrate cell death.
Thank you very much for this good advice. Cell Viability assays have now been performed and are now integrated into the manuscript as the new Supplementary material 1b and 1e.
- Figure 1 is not so important and can be put in supplemental data. Figure 1 should be something very important and striking. The current figure 1 is sort of disappointing. The authors should consider re-organize the figures.
Thank you very much for this good suggestion. Figure 1 has been moved to the supplementary data as the new Supplementary Figure 1.
- According to the illustration in figure 2A. fw-eRL plus rv primers can also amplify uncut template. Therefore, eRL band is not specifically derived from cut eRL. The authors should explain the specificity of this PCR system. Otherwise the data obtained from this assay is highly shaky.
Thank you for this helpful advice. An explanation of the method and a proof of its specificity have been added (please compare Figure 1a and results section 174 – 182). We agree that also from uncut ITS2-RNA, the shorter eRL-fragment can be synthesized, however, this only occurs at a lesser extent, approximately at a 7:1 ratio regarding band intensities in an agarose gel. This is because if the longer, uncut RNA sequence is present and the fw-uncut primer can bind, this leads to displacement of the fw-eRL primer, resulting in preferential amplification of the uncut template (Figure 1a, right panel, mock sample). Generation of the eRL leads exclusively to synthesis of the shorter sequence (Figure 1a, right panel, digested sample). In the case of intracellular eRL generation, the ratio between uncut sequences and eRL sequences is consequently reversed (compare Figure 1b, line H).
- The legend of Figure 2 requires a topic/summary sentence.
Thank you very much for this conscientious observation. This information has been included (now Figure 1).
- Figure 3b. eRL is missing. the author need to describe how they stimulated the cells with the RNAs. Is it transfection? How much RNAs? How long?
Thank you for this useful advice. eRL data have been integrated into the new Figure 2b. Regarding the stimulation procedure, we would like to refer to the materials and methods section, in particular to the subchapters “Luciferase assay” and “PBMC Stimulation”. RNA ligand concentration for Supplementary Figure 2B has been included in the figure legend.
- Topic sentence if missing in the legend of Figure 3.
Thank you very much for this conscientious observation. This information has been included (now Figure 2).
- The data from Figure 3 is basically well-known because it contains genomic RNA of IAV. There is no novelty in figure 3.
Thank you very much for your patience. We completely agree with you in this regard. It just seemed important to us for the rest of the paper to verify that the RNA from our experimental setups also exhibits these properties. Here, the experiments with the PBMCs are particularly important to us, as they are the "original system", which was not used in the further course of the project due to its lack of specificity by expressing further receptors of innate immunity.
- Line 190. (CIP)removes -> (CIP) removes.
Thank you for carefully proofreading our manuscript. The space has been inserted.
- Mock treatment controls are missing in figure 4.
Thank you very much for this valuable comment. Corresponding data have been integrated into the new figure 3d and Supplementary Figure 4.
- In figure 4, W/O CIP vs +CIP should be statistically analyzed and discussed.
Thank you very much for this helpful advice. Corresponding sections have been included in the manuscript and in the new Figure 3. Of note, immunostimulation by RNA extracted from IAV infected cells was significantly decreased after CIP treatment. The decrease of stimulatory effects in CIP-treated RNA compared to the untreated RNA reflects the dephosphorylation of IAV genomes. However, we still saw a significant upregulation of IFN beta induction by CIP-treated RNA from infected cells compared to RNA from mock-infected cells. Consequently, immunostimulation in IAV-infected cells does not depend solely on 5'ppp RNA and thus does not depend solely on IAV genomes.
- Figure 4d is the most valuable data in figure 4. However, mock-infected RNA is a required control to demonstrate the stimulus was derived from infection rather than background. VSV infected cell RNA should also be included in figure 4.
Thank you very much for this good suggestion. Following your suggestion, we also tested the VSV-induced RNA in immunological assays now (please see supplementary Figure A4). We found that VSV-induced RNA species also activate RIG-I in an eRL-like manner. We suspect that another endogenous RNA ligand, other than the sequence we described, might contribute to innate immune activation.
- Mock treatment controls are missing in figure 5 .
Thank you for this helpful suggestion. These data have been integrated into the new figure 4.
- VSV infected sample should also be used in figure 5 to demonstrate no eRL was induced.
Thank you for this helpful suggestion. These data have been integrated into the new supplementary Figure A5.

Reviewer 2 Report
The work shows many techniques and is well structured, despite this I have some doubts for which I think the work needs to be revised.
In general, why there are so few authors of such technical work, and only three authors?
Also, the HCl treatment, despite the previous literature, I find to be very aggressive. Have viability/death tests been done? They should be put in the work.
In the materials and methods, Western blot is mentioned but no protein data is given in any figure in the work. Nor of image. How so?
I would also evaluate a revision of the English and the possibility of adding a graphical abstract that helps the reader in understanding the work that is certainly complex.
Author Response
Thank you very much for your positive and helpful feedback. We have processed the points as indicated below. We think that the experiments and changes you suggested greatly improved the quality of the publication, and we hope that you now find our work as compelling as we do.
In general, why there are so few authors of such technical work, and only three authors?
Thank you very much for your patience and helpful comments. In fact, only three people contributed to this work, namely Julia Steinberg (PhD candidate), Timo Wadenpohl (technician) and Stephanie Jung (Group Leader). Our group “Cellular Virology” is just getting started, since Stephanie Jung took up her position as group leader in mid-October 2020, and only consists of these three people so far. The present work was greatly aided by the fact that it builds on our previous paper "A ribosomal RNA fragment with 2 ,3 -cyclic phosphate and GTP-binding activity acts as RIG-I ligand" published with Stefan Bauer in Marburg, so essential techniques were already known. However, the present data were generated exclusively by the "Cellular Virology" team in 2021.
Also, the HCl treatment, despite the previous literature, I find to be very aggressive. Have viability/death tests been done? They should be put in the work.
Thank you very much for this helpful concern. Indeed, we believe that neutralization of RNA ligand solution with NaOH helps to minimize cytotoxic effects of HCl treatment. Viability assays were performed and included into the manuscript as the new Figure A6.
In the materials and methods, Western blot is mentioned but no protein data is given in any figure in the work. Nor of image. How so?
We apologize that the separate Western Blot upload got lost in the submission process. Please find the Western Blot analysis of IAV M2 protein, VSV glycoprotein and housekeeping gene expression in the new supplementary Figure 1.
I would also evaluate a revision of the English and the possibility of adding a graphical abstract that helps the reader in understanding the work that is certainly complex.
Thank you very much for these very good ideas. The manuscript has been revised by a native-English speaking colleague. A graphical abstract has been included as the new Figure 5.
Reviewer 3 Report
Review of: Viruses-1314099
Title: “The endogenous RIG-I Ligand is generated in Influenza A-virus infected cells”
by: Julia Steinberg, Timo Wadenpohl and Stephanie Jung
This is a reasonably well-written paper, that tries to elucidate whether the eRL (an immune-stimulatory RNA fragment, generated upon viral infection, known to function as an endogenous RIG-I ligand) is also generated in Influenza A (IAV) and vesicular stomatitis virus (VSV) infected cells. The work is heavily based upon previous work of the last author, Stephanie Jung (Jung S, von Thülen T, Yang I, Laukemper V, Rupf B, Janga H, Panagiotidis G-D, Schoen A, Nicolai M, Schulte LN, Obermann H-L, Weber F, Kaufmann A, Bauer S. 2020. A ribosomal RNA fragment with 2,3-cyclic phosphate and GTP-binding activity acts as RIG-I ligand. Nucleic Acids Research 48:10397-10412). The broad aim is to investigate if, and under which conditions, eRLs are generated in viral infections. Generation of eRL and its effects were demonstrated using both in vitro and reporter systems. The manuscript reflects a nice build-up of logical experiments to come to further insights regarding the process of specific cellular antiviral mechanisms. The main conclusion: eRL is indeed generated during IAV infections, but the authors were unable to detect eRL generation in VSV infected cells. Overall, I liked the paper. It is to the point, adequate and concise, though really limited in scope. The discussion is more than adequate as well.
However, the manuscript is not completely error free (yet).
I thus have some minor quibbles with the authors, mostly regarding language (but not restricted to that category).
Minor Points (Majority: language problems/suggestions):
Line 7: “…but also…”; “…but it also…”
Line 56/57: “at investigating if and under which conditions the eRL”; “at investigating if, and under which, conditions the eRL”
Line 153: “in profoundly detectable eRL induction”; “in clearly detectable eRL induction”
Line 165: “experiments, validated to mock-treated” (that is incorrect); “experiments, compared to mock-treated”
Figure 3b: where is the “eRL” column? I know that 3C shows induction, but in that case we are looking at the luciferase reporter outcome… Now the reader might start to surmise that the experiment was left out because its result was contradictory.
Line 189/191: “Whereas calf intestine phosphatase (CIP)removes 5’ phosphorylation of classical RIG-I ligands, such as IAV genomes, is the 2’3’ cyclic phosphate of the eRL resistant to CIP treatment (19).”; “Whereas calf intestine phosphatase (CIP)removes 5’ phosphorylation of classical RIG-I ligands, such as IAV genomes, the 2’3’ cyclic phosphate of the eRL is resistant to CIP treatment (19).”
Line 232: “what extend”; “what extent” (a common mistake: the noun is “extent”, the verb is “to extend”).
In the next part, the authors use inappropriate styles (we are not recounting a fairy-tale here). Thus:
Line 235: “At first, we”; “First, we”
Line 244: “In the following, we”; “Next, we”
Line 246/247: “In the further course, we showed”; “Hereupon we showed”
Line 263/264: “ligands plays a considerable in IAV-mediated”; “ligands plays an important role in IAV-mediated”
Line 266: “keyrole”; “key role”
Line 273: “OAS”; Don’t forget the article is also for non-experts: explain that its name comes from “2'-5' oligoadenylate synthetases”….
Line 291: “shown too, that processing”; “shown too that processing”
Author Response
Thank you very much for your positive feedback and your conscientious comments. We have processed the points as indicated below. We think that the experiments and changes you suggested greatly improved the quality of the publication, and we hope that you now find our work as compelling as we do.
Minor Points (Majority: language problems/suggestions):
Line 7: “…but also…”; “…but it also…”
The sentence has been corrected in this way.
Line 56/57: “at investigating if and under which conditions the eRL”; “at investigating if, and under which, conditions the eRL”
Thank you very much for your help, also with the English language. The commas were integrated into the sentence.
Line 153: “in profoundly detectable eRL induction”; “in clearly detectable eRL induction”
Thank you for your help. We have now adopted this better worded sentence.
Line 165: “experiments, validated to mock-treated” (that is incorrect); “experiments, compared to mock-treated”
Thank you for your thoughtful correction. The change has been applied.
Figure 3b: where is the “eRL” column? I know that 3C shows induction, but in that case we are looking at the luciferase reporter outcome… Now the reader might start to surmise that the experiment was left out because its result was contradictory.
Thank you for this good advice. eRL data have been integrated into the new Figure 2b.
Line 189/191: “Whereas calf intestine phosphatase (CIP)removes 5’ phosphorylation of classical RIG-I ligands, such as IAV genomes, is the 2’3’ cyclic phosphate of the eRL resistant to CIP treatment (19).”; “Whereas calf intestine phosphatase (CIP)removes 5’ phosphorylation of classical RIG-I ligands, such as IAV genomes, the 2’3’ cyclic phosphate of the eRL is resistant to CIP treatment (19).”
Thank you very much for your good advices. The sentence has been replaced.
Line 232: “what extend”; “what extent” (a common mistake: the noun is “extent”, the verb is “to extend”).
Thank you for this good explanation, I should remember that. The change has been applied.
In the next part, the authors use inappropriate styles (we are not recounting a fairy-tale here). Thus:
Thank you very much, I completely agree with you. I have changed the sentences in this way.
Line 235: “At first, we”; “First, we”
Line 244: “In the following, we”; “Next, we”
Line 246/247: “In the further course, we showed”; “Hereupon we showed”
Line 263/264: “ligands plays a considerable in IAV-mediated”; “ligands plays an important role in IAV-mediated”
Thank you for your attentive corrections, this and the next change have also been applied.
Line 266: “keyrole”; “key role”
Line 273: “OAS”; Don’t forget the article is also for non-experts: explain that its name comes from “2'-5' oligoadenylate synthetases”….
Thank you for this helpful comment, the explanation has been integrated.
Line 291: “shown too, that processing”; “shown too that processing”
Thank you for all your help, the comma has been integrated.

Round 2
Reviewer 1 Report
If the authors admit that the shorter band can be derived from uncut template, annotating the shorter band as eRL is kind of misleading. After all, the shorter band can be purely derived from uncut template and has nothing to do with eRL.
Author Response
Thank you very much for your conscientious feedback. We would like to refer to Figure 2C and Figure 4 of our Nucleic Acid Research Paper from 2020 as attached. We could also already convincingly show by Northern Blot the eRL is generated and that this correlates with the results of the PCR. In this publication here, we used the same primers and conditions as in the previous publication.

Reviewer 2 Report
I wondered about the number of people in the work because there are so many results and techniques applied and I was amazed, the weaknesses of the work show a little care in the final review of the work before the submission and you have to be very careful about this. However, now the authors have corrected as requested so the work is now ready to be published. I appreciate the new and young groups so good luck.
Author Response
Thank you very much for your good wishes, and your nice and positive feedback! I’m also really grateful that I could recruit such great colleagues as Julia and Timo, with whom a publication is possible in such a short time.
All the best
Stephanie Jung